# Time distribution for pharmacists conducting a randomized controlled trial—An observational time and motion study

**Kjerstin Havnes**[1]*, **Elin C. Lehnbom**[1,2], **Scott R. Walter**[3], **Beate H. Garcia**[1], **Kjell H. Halvorsen**[1]

**1** Department of Pharmacy, UiT the Arctic University of Norway, Tromsoe, Norway, **2** Faculty of Health and Life Sciences, Department of Health and Caring Science, Linnæus University, Kalmar, Sweden, **3** Faculty of Medicine Health and Human Sciences, Centre for Health Systems and Safety Research, Australian Institute of Health Innovation, Macquarie University, Sydney, Australia

* Kjerstin.havnes@uit.no

**Data Availability Statement:** All data are publicly available at the UiT Open Research Database (https://doi.org/10.18710/PWFB8M).

## Abstract

### Introduction

An expected future increase in older adults will demand changes in health care delivery, making development, implementation and evaluation of new health care models essential. The rationale for political decision-making concerning the implementation and application of interventions in health care should include cost estimations, specifically those involving clinical interventions. To provide such data knowledge of time spent on the intervention is imperative. Time and motion methodology is suitable to quantify health care personnel's time distribution.

### Aim

To investigate the time distribution for pharmacists conducting a randomized controlled trial (RCT) implementing a clinical intervention.

### Materials and methods

The setting was an RCT with a 5-step pharmacist-intervention in collaboration with the interdisciplinary team in a geriatric ward. Two pharmacists were involved in the trial during the observation period. Pharmacist activities, classified as RCT-tasks (intervention or administrative), non-RCT tasks and social/breaks, were recorded applying the Work Observation Method By Activity Timing methodology, enabling recording of predefined work tasks as well as interruptions and multitasking. One observer collected data over eight weeks.

### Results

In total, 109.1 hours were observed resulting in 110.2 hours total task time, including multitasking. RCT tasks comprised 85.4% of the total observed time, and nearly 60% of the RCT time was spent on intervention tasks. Medication reviews was the most time consuming task, accounting for 32% of the observed time. The clinical pharmacists spent 14% of the

**Funding:** The author(s) received no specific funding for this work.

**Competing interests:** The authors have declared that no competing interests exist.

intervention time communicating verbally, mainly with patients and healthcare professionals.

## Conclusion

During the RCT, the clinical pharmacists spent about half their time performing the actual intervention. Consequently, costs for providing such a clinical pharmacist service should reflect actual time spent; otherwise, we may risk overestimating theoretical costs.

## Introduction

The predicted future increase in older adults will demand changes to the way health care is delivered. An expansion and/or reallocation of health care spending and human resources across different care settings seems inevitable [1,2]. Consequently, evaluating practice models for health care services, including different health care team collaborations and compositions, has become increasingly relevant.

The randomized controlled trial (RCT) design is suited to evaluate complex health interventions and organizational changes by reducing biases, enabling comparisons of new work models with standard care, and presenting data on defined outcomes [3–6]. In addition, the RCT has substantial potential to provide information on costs related to carrying out interventions [7,8]. To guide the economic evaluation of health interventions The Consolidated Health Economic Evaluation Reporting Standards (CHEERS) was released in 2013 [9]. Ten journals published the standard simultaneously, indicating consensus regarding the need for such guidelines. Nevertheless, a later literature review aiming to support general RCT-planning found only partial reporting of costs in the studies assessed [10]. Similar findings have been reported in Norway [7].

Cost reporting is highly relevant for studies of clinical pharmacy services. Evaluations of the economic implications of such services have shown consistent shortcomings [11–17]. Although most studies present beneficial cost-effectiveness, the variability of study design, outcomes and methods for cost-estimations are substantial, making it difficult to achieve general conclusions. Few studies report full economic evaluations, and if they do the quality is often deemed poor, for example input costs are rarely included in the evaluations [13,16]. Even recent studies of clinical pharmacy services do not adhere to the CHEERS criteria [14].

Health care personnel expenses in clinical trials are often referred to as direct costs (related to an intervention, or the disease) as opposed to the indirect costs (related to loss of productivity, patient perspective) [18,19]. To provide more powerful evidence supporting clinical pharmacy services, a more detailed overview of intervention costs of clinical pharmacy services has been called for [14,16]. Reliable cost estimates of health care interventions depend upon the differentiation between costs of performing the intervention and costs associated with running the study [18]. However, to our knowledge, there are no studies determining how clinical pharmacists spend their time conducting an RCT. This should be of interest for health policy decision-makers responsible for prioritizing and allocating resources to provide efficient and equitable health care, since poorly designed or non-existent cost-estimates may give an inaccurate basis for decision-makers regarding whether to implement interventions or not.

The aim of this study was to investigate the time distribution for pharmacists delivering the clinical pharmacist intervention while also operating day-to-day administrative responsibilities in an RCT.

## Materials and methods

### Design and setting

We conducted a time and motion study of clinical pharmacists delivering a clinical pharmacy service while performing day-to-day administrative responsibilities in the IMMENSE study [20]. The IMMENSE-study is a two-armed RCT in an acute geriatric ward at the University Hospital of North Norway. The control arm comprised standard care provided by an interdisciplinary team consisting of physicians, nurses, physio-, speech-, and occupational therapists. In the intervention arm, the clinical pharmacists were included in the interdisciplinary team carrying out the following clinical tasks: medication reconciliation at admission, medication review during the hospital stay, patient counselling during hospital stay and at discharge, medication reconciliation at discharge including documentation of a structured medication list, and finally a telephone call to the patients' primary care physician within two weeks after discharge to discuss medication changes and treatment plan, see Fig 1.

During data collection two clinical pharmacists shared a 100% position, delivering the intervention, recruiting and randomizing patients, as well as collecting and documenting study relevant information. They were at the ward weekdays between 8.00 a.m. and 3.30 p.m. (37.5 hours/week).

### The observation tool

The time and motion methodology is well suited to collect information about how health care personnel allocate time on different tasks and activities. Such data are widely used to describe and improve production efficiencies in health care systems and commonly used to evaluate costs of provided care [21–23].

For the time and motion observations we applied the validated Work Observation Method of Activity Timing (WOMBAT) methodology [24,25]. WOMBAT comprises continuous observation of workflow, performed by an external observer. Pharmacist activities were recorded by using a tablet (Samsung Galaxy Tab S2) with the validated WOMBAT software downloaded. The software allows for continuous recording of time in multiple dimensions [24,25]. In this study, we recorded the following four dimensions: *what* (task performed), *who* (with whom the observed pharmacist interacts), *how* (by phone, face-to-face, electronic journal and transit) and *where* (the location of the task). The *what*- dimension was mandatory and always recorded, while the other three dimensions were optional [24–26]. When a task with any multiselection of dimension sub-categories was recorded, the software automatically time stamped the interval of each task together with any additional selection. Interruptions (i.e. response to an external stimuli which causes the observed pharmacist to change task) and multitasking (i.e. when two or more tasks are performed simultaneously) are also recorded with time stamps [25].

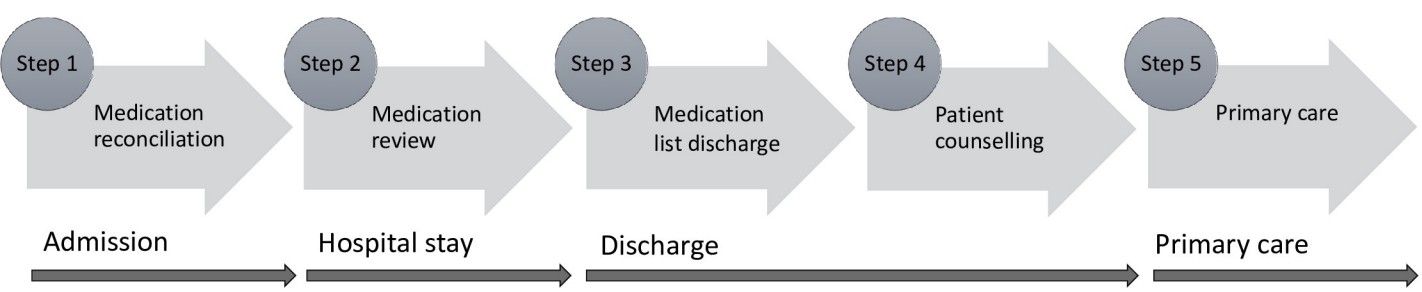

**Fig 1. The RCT intervention.**

**Table 1. Final version of the WOMBAT observation categories with mutually exclusive definitions.**

| Dimension | Category | Work task | Definition |
|---|---|---|---|
| **What** | RCT | Medication reconciliation (i) | Perform medication reconciliation: information gathering, interviews regarding the patients' medication use, communicate findings. *Excludes*: copy charts, plot in study database, write in electronic journal. |
| | | Medication review (i) | Reviewing inclusion patients' medication use: symptom score, documentation, assessing patient journals, databases and guidelines. *Excludes*: plot in the study-database. |
| | | Counselling (i) | Counsel intervention-patients: advice about medicine/health-related issues. *Excludes*: conversation about medication list at discharge. |
| | | Discharge Medication list (i) | Preparing medication lists at discharge: Clarifying medication use, work with lists in electronic journal, guide physicians using the lists. *Excludes*: when lists have status "finished" in electronic journal and is printed. |
| | | Discharge intervention (i) | Review of medication lists at discharge: communication with primary care/patient regarding medication lists, -use and prescriptions. *Excludes*: communication about medication use at admittance. |
| | | Information RCT (a) | Information of RCT to collaborating professions, overlapping RCT-information between study-pharmacists. *Excludes*: explain the study during inclusion. |
| | | Map patient flow (a) | Identifying patient flow: verification of eligibility and ability to consent. *Excludes*: monitor included patients. |
| | | Inclusion RCT (a) | Enrolling patients: preparing forms, providing RCT information to patient/next of kin, getting consent, randomization and documentation. *Excludes*: classification of patient status. |
| | | Registration/ organization (a) | Organizing RCT and registration in study specific systems: printing data, copy charts, filing, and registration in study database. *Excludes*: work with inclusion forms, log data in patient journals. |
| | Social /breaks | Social/ breaks | Social interaction and breaks: lunch, get drinks, restroom breaks, conversations, private phone calls. |
| | Non-RCT | Clinical, non-study related | Work regarding patients not included in the RCT. |
| | | Other | Other tasks not already defined, such as turning the computer on/off, retrieving work documents, hospital pharmacy meeting activities. |
| **How** | | | Medium of action: phone (= verbal communication), face-to-face (= verbal communication), electronic journal and transit. |
| **Who** | | | With whom the pharmacist interacts: physician, patient, nurse, next of kin, home care services, pharmacist and other. |
| **Where** | | | Location of performed activities. |

(i) = intervention task, (a) = administrative task.

The categories within each dimension were based on knowledge about the RCT and ward activities. Pharmacist activities (*what*) were differentiated and classified as either 1) RCT-tasks, 2) non-RCT-tasks or 3) social activities/breaks. The latter comprised activities such as not work-related conversations, toilet and lunch breaks, and were not differentiated further. The RCT category, however, was divided into two subgroups: intervention and administration. The intervention tasks evolved from the clinical intervention described in Fig 1, while the administrative tasks were related to conducting the RCT, e.g. including patients, collecting and plotting study data.

The external observer (EL), having extensive WOMBAT experience, and a clinical pharmacist previously involved in the RCT (KH), conducted preliminary observations and piloted the WOMBAT software tool. See Table 1 for the final version of the WOMBAT observation categories, including mutually exclusive definitions describing inclusion and exclusion criteria for each task category and dimension.

## Data collection

The external observer, instructed not to interact with either pharmacists, patients or ward staff, observed the clinical pharmacists during an eight-week period and recorded all the tasks they performed during each observation session. Throughout this period 36 patients were asked to participate in the RCT. Of these 30 patients accepted, where 16 were randomized to

the intervention group. A total of 106 patients were admitted during the observation period, of whom 66 were eligible for inclusion. The observation schedule was planned to ensure proportional observation during the pharmacists' working hours. We divided daily working hours into four intervals and restricted each data collection period to a maximum of 1 hour 55 minutes. To avoid observation fatigue we only allowed two data collection periods daily, and never consecutively. The pharmacists were informed about the scheduled observations. After the final observation session, the external observer and the observed pharmacists participated in a debrief (led by KH) to collect information about their experiences of both observing and being observed.

## Statistical analysis

Data from the WOMBAT software was downloaded as a comma-separated value (CSV) file and analyzed with the Statistical Analysis Software (SAS) system for Windows, version 9.4. Time spent on each task category was expressed as a proportion of total observation time. Total task time was expressed as time spent on multiple tasks. Interruption rates were generated for relevant categories along with the proportion of time spent multitasking. Furthermore, we generated proportions of time in sub-categories from the defined time-categories and from verbal interaction. We calculated corresponding 95% confidence intervals (CIs,) using a simple bootstrap approach where the 2.5[th] and 97.5[th] percentiles of 1000 resampled measures were used as the lower and upper confidence limits.

## Ethics approval

As no sensitive information was collected in the study no ethical approval was necessary in accordance with national guidelines in Norway (The Regional committee for medical and health research ethics, Norway, reference number 2017/685). However, both pharmacists consented to be observed, confirmed in writing. We also informed the health care personnel at the ward of the study.

## Results

The pharmacists were observed for a total observation time (TOT) of 109.1 hours, giving 110.2 hours total task time due to observed multitasking. The time distribution and proportions of TOT are presented in Table 2.

The clinical pharmacists spent 85.4%, 9.7% and 5.8% of the total observation time on RCT-tasks, non-RCT-tasks and social activities/breaks, respectively. Undertaking RCT-tasks the pharmacists spent 59% of the time (55.3 hours) carrying out the clinical intervention and spent 41% of this time on administrative tasks. While carrying out the intervention most time was spent on medication review (63%) followed by provision of a discharge medication list (19%). Within the administrative tasks most time was allocated to study registration/organization. Of non-RCT tasks, 9.4 hours (89%) was spent on "Other".

The pharmacists were interrupted 1.5 (95% CI 1.1–1.8) times per hour when occupied with the intervention, slightly higher than the observed average interruption rate of 1.3 per hour (95% CI 1.1, 1.5). During the observations, about 1% of the total time was recorded as multitasking, with 250 instances in total. The multitasking presented mostly within the social/breaks category (191 instances, 13% of the social/breaks time). The pharmacists communicated verbally 18.8% of the total time observed. Table 3 displays the communication time distribution.

Social/breaks together with non-RCT tasks consisted of more than 50% verbal communication. However, most of the verbal communication (59.7%) took place during RCT-activities,

**Table 2. Distribution of time for all tasks.**

| Category | Observed time (h) | Percentage of TOT | Sub-category | Work task | Task time (h) | Proportion of TOT (95% CI) | Proportion of intervention specific task time (95% CI) |
|---|---|---|---|---|---|---|---|
| RCT | 93.3 | 85.5 | Intervention (55.3 h) | Medication review | 34.9 | 32.0 (28.4–36.8) | 63.2 (55.3–71.4) |
| | | | | Discharge medication list | 10.5 | 9.6 (7.7–11.9) | 19.0 (15.1–23.2) |
| | | | | Medication reconciliation | 7.4 | 6.7 (5.2–8.7) | 13.3 (10.0–17.0) |
| | | | | Discharge intervention | 2.2 | 2.0 (1.2–3.0) | 3.9 (2.3–5.8) |
| | | | | Counselling | 0.3 | 0.3 (0.1–0.7) | 0.6 (0.1–1.2) |
| | | | Administration (38.0h) | Registration/ organization | 25.9 | 23.7 (20.8–27.4) | n/a |
| | | | | Map patient flow | 5.6 | 5.1 (4.2–6.4) | n/a |
| | | | | Inclusion RCT | 4.9 | 4.5 (3.6–5.6) | n/a |
| | | | | Information RCT | 1.6 | 1.5 (0.8–2.6) | n/a |
| Non-RCT | 10.6 | 9.7 | | Other | 9.4 | 8.6 (5.8–12.9) | n/a |
| | | | | Clinical, non-study related | 1.2 | 1.1 (0.6–1.6) | n/a |
| Social/ breaks | 6.3 | 5.8 | | Social/breaks | 6.3 | 5.8 (4.7–7.3) | n/a |

Tasks are presented as proportions of total observational time (TOT) and intervention tasks are also specified as proportions of RCT-intervention task time.

h = hours, CI = confidence interval.

of this 61% was recorded when executing the intervention, which accounted for 14% of the intervention specific time, mainly with physicians (36.8%) and patients (29.8%), see Fig 2.

## Discussion

The current study is the first to explore time distribution for pharmacists delivering a clinical pharmacist intervention while also operating day-to-day administrative responsibilities in an RCT. The pharmacists spent a considerable proportion of the observed time performing RCT-administrative tasks, which was expected since RCTs are known to be time-consuming and require extensive administration [27]. Time spent on study administration should be of little importance for key stakeholders assessing whether or not to implement new health care services; thus, the RCT-administrative proportion of the pharmacist time should be omitted from future economic assessment of the intervention. However, these findings may provide valuable information to health care researchers planning similar studies.

The majority of the intervention time (63%) was spent on medication reviews. WOMBAT-studies of real-life clinical pharmacy practice in a children's ward and public hospitals report

**Table 3. Distribution of the verbal communication time.**

| Category | Time (h) | % of total communication time (CI 95%) | RCT subcategories | Time (h) | % of RCT specific communication time (CI 95%) |
|---|---|---|---|---|---|
| RCT | 12.2 | 59.7 (50.6, 70.5) | Intervention | 7,5 | 61.0 (49.0, 74.1) |
| | | | Administration | 4.8 | 39.0 (28.7, 50.6) |
| Non-RCT | 4.8 | 23.6 (8.1, 45.4) | n/a | | |
| Social /breaks | 3.4 | 16.8 (11.5, 23.1) | n/a | | |

h = hours, CI = confidence interval, RCT = randomized controlled trial.

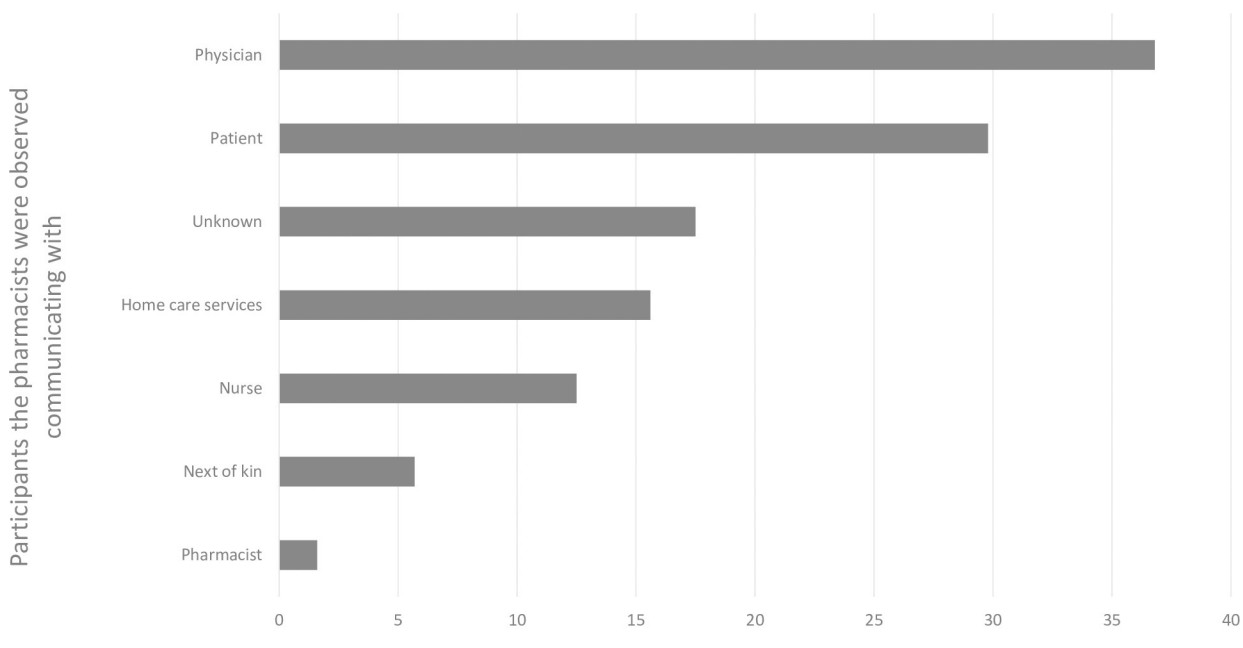

**Fig 2. Pharmacist communication during the intervention-specific time (n = 7.1 hours).**

that about 30% of the observed time is spent on performing medication review [26,28]. Although direct comparison is difficult due to the differences in settings, category design and definitions, the proportion appears relatively high and may reflect that the pharmacists valued conducting medication reviews above administrative tasks. Other explanations include that few patients were available for inclusion, no time limits or deadlines for tasks completion had been set [20], and medication reviews were usually presented to physicians the day after inclusion. Also, medication reviews in frail elderly patients with complex medication regimens requires a thorough clinical assessment making them time-consuming. In addition, the pharmacists might have been more thorough in the RCT, knowing that the end results of the intervention would influence the outcomes of the trial.

In contrast, the proportions of time spent on medication reconciliation and the provision of drug information to patients at discharge were low. Both these tasks have rather explicit endpoints and were conducted typically in a semi-structured form with checklists in the RCT.

The total time spent working clinically *per patient* was substantial in the observed study. During the observations, on average 3.5 hours was spent performing clinical tasks per intervention patient. Studies on optimal length for pharmacists performing clinical services in hospitals are scant, but two Swedish trial studies estimate that pharmacists conduct their clinical tasks spending around 30–65 minutes per patient [29,30]. Both these estimates differ notably from our findings to a point where the feasibility of the intervention can be questioned, unless health economic studies of the intervention demonstrate differently.

When working with intervention tasks the pharmacists communicated verbally 14% of the time, mainly with physicians and/or patients. A cross-country WOMBAT study measuring the impact of electronic medication management systems on the work of hospital pharmacists in Australia and England reported above 30% of the observed time spent on tasks implying communication pre and post implementation[28]. Lehnbom et al. reported 25% communication, including medication discussion, in a pediatric hospital setting [26]. Since our study recruited

intervention and control patients from the same ward, and clinical pharmacists were not allowed to engage in matters regarding control patients (to minimize bias), the time spent communicating seems comparable [20]. Even though such behavior might be beneficial when running an RCT, it deviates from the role expected of clinicians working in interprofessional teams [31], adding to the differences between the observed setting and clinical practice.

## Implications of findings

By quantifying the time clinical pharmacists spend on different tasks we enable estimations of pharmacist costs in future economic evaluation of the provided health care service. Our presentation of the clinical pharmacists' time distribution is highly relevant in the evaluation of the observed study's outcomes, facilitating a detailed overview of pharmacist costs in accordance with the CHEERS criteria [9]. But more interestingly, the study demonstrates inevitable transferability issues, pointing to limitations by using results from a fixed study design to provide sound cost estimates of a real-life clinical service. On the other hand, our results support the importance of conducting accurate planning. They could also be used to recalibrate workflow efficiency of similar services by suggesting appropriate time allocation to individual tasks.

## Strengths and limitations

The main strength of this study is the observational category framework in combination with the WOMBAT software, enabling exact time registration of activities, similar to what has been described when measuring time distribution of physicians, nurses and pharmacists in various hospital settings [22,26,32–36]. The comprehensive development of categories based on the RCT structure reflects the workflow in a multidimensional clinical study, and made measurement an easy task. Moreover, all observations took place in the beginning of study year three of the RCT, ensuring that routines were well-established and thereby less prone to change due to outside factors. The use of an experienced WOMBAT observer (EL) [26,28,37] is another strength of the study, most likely favoring precision but not the accuracy of the measurements. Other time and motion studies have reported acceptable inter-observer reliability above 85% consensus, indicating that some variations between recordings are expected [24,25]. The conducted debrief revealed that some of the nurses hesitated to contact the clinical pharmacists when data was recorded, probably resulting in an underestimation of interruptions, multitasking, and possibly affecting verbal communication time. Neither can we disregard the possibility that having an observer present may have affected the participating pharmacists [38].

## Conclusion

The clinical pharmacists spent approximately the same amount of time performing clinical and administrative tasks when running an RCT. The theoretical costs of the intervention would be grossly over-estimated if clinical and administrative tasks were not differentiated.

The pharmacists devoted substantial time to performing medication reviews, which most likely reflect the research setting rather than the standard clinical practice behavior. This study highlights the importance of assessing time spent on different tasks when performing clinical services and that time and motion studies of clinical pharmacists in real-life geriatric ward settings are warranted.

## Acknowledgments

The authors would like to thank the participating pharmacists for their contribution to the study.

## Author Contributions

**Conceptualization:** Kjerstin Havnes, Elin C. Lehnbom, Beate H. Garcia, Kjell H. Halvorsen.

**Data curation:** Scott R. Walter.

**Formal analysis:** Scott R. Walter.

**Investigation:** Kjerstin Havnes, Elin C. Lehnbom.

**Methodology:** Kjerstin Havnes, Elin C. Lehnbom.

**Project administration:** Kjerstin Havnes.

**Supervision:** Kjell H. Halvorsen.

**Validation:** Kjerstin Havnes, Elin C. Lehnbom, Scott R. Walter, Kjell H. Halvorsen.

**Visualization:** Kjerstin Havnes, Elin C. Lehnbom, Scott R. Walter, Beate H. Garcia, Kjell H. Halvorsen.

**Writing – original draft:** Kjerstin Havnes, Elin C. Lehnbom, Scott R. Walter, Beate H. Garcia, Kjell H. Halvorsen.

**Writing – review & editing:** Kjerstin Havnes, Elin C. Lehnbom, Scott R. Walter, Beate H. Garcia, Kjell H. Halvorsen.

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
