## [Decision Letter · Decision Letter 0]

2 Mar 2021

PONE-D-21-02659

Time distribution for pharmacists conducting a randomized controlled trial– an observational time and motion study

PLOS ONE

Dear Dr. Havnes,

Thank you for submitting your manuscript to PLOS ONE. After careful consideration, we feel that it has merit but needs some revision. Therefore, we invite you to submit a revised version of the manuscript that addresses the points raised during the review process.

We look forward to receiving your revised manuscript.

Kind regards,

Binaya Sapkota, PharmD

Academic Editor

PLOS ONE

Journal Requirements:

Additional Editor Comments:

Thank you for giving us the opportunity to review and consider your manuscript. It has been reviewed and the referees’ comments are listed below. Please address the concerns raised by the reviewers point-by-point before we consider it further.

Reviewers' comments:

Reviewer's Responses to Questions

**Comments to the Author**

1. Is the manuscript technically sound, and do the data support the conclusions?

Reviewer #1: Yes

Reviewer #2: Yes

2. Has the statistical analysis been performed appropriately and rigorously? 

Reviewer #1: Yes

Reviewer #2: Yes

3. Have the authors made all data underlying the findings in their manuscript fully available?

Reviewer #1: Yes

Reviewer #2: Yes

4. Is the manuscript presented in an intelligible fashion and written in standard English?

Reviewer #1: Yes

Reviewer #2: Yes

5. Review Comments to the Author

Reviewer #1: Thank you for the opportunity to review this paper. I was very interested in the use of the WOMBAT methodology and I think this paper will be useful for other researchers in this area.

My comments are very minor - more grammatical than methodological.

General Comments:

Referencing: numbered referencing styles require the numbers to be placed outside punctuation marks with in-text citing

Punctuation: commas in wrong places – punctuation marks need checking

Abstract:

Line 25: suggest changing ‘performing’ to ‘involving’ clinical interactions

Line 26: rather than ‘time allocation’ would it not be ‘time spent’ on a clinical intervention?

Line 43: suggest change ‘allocation’ to time ‘spent’

Introduction

Line 47: suggest changing ‘relocation’ to ‘allocation’

Line 53: change ‘present’ to ‘presenting’ and remove ‘a’ in front of ‘substantial’

Line 56: ‘released’ instead of ‘issued’

Line 84: change ‘operating’ to ‘performing’

Line 110/111: suggest rewording the sentence about the dimensions being recorded and when – suggest list the four dimensions and then in a new sentence tell us when the various dimensions were recorded.

Results

Line 175: suggest rewording to ‘spent 41% of this time on administrative tasks’.

Discussion

Line 196: One sentence is not a paragraph – remove break between the first two sentences.

Line 201: “Time spent on study administration is of little importance for key stakeholders assessing whether or not to implement new health care services” – who says? Need to reference this statement.

Line 221: might be useful to repeat the results for the time spent clinically per patient in this study to save reader flicking back through the results section

Conclusion

Line 269: “Without reduction of the costs based on the actual time distribution” – I think what you are trying to say is that without pulling the time spent on administrative tasks out of the costings so that only the actual time spent conducting the clinical intervention is costed, the theoretical costs of the intervention will be over-estimated. Therefore, suggest rewording for clarification.

Reviewer #2: This was an excellent study; well planned and executed. The reasons for the study were clearly written. The roles of pharmacists in RCTs and time spent are clearly explained. The authors had also provided the implications. I also appreciate the points on the strengths and limitations of the study. The concluding remarks were consistent with the study objectives and findings.

6. PLOS authors have the option to publish the peer review history of their article (what does this mean?). If published, this will include your full peer review and any attached files.

Reviewer #1: No

Reviewer #2: **Yes: **Mohamed Izham Mohamed Ibrahim

---

## [Author Response · Author response to Decision Letter 0]

17 Mar 2021

Thank you for the feedback to our manuscript. We have adressed all comments in the uploaded file Response to reviewers.

---

## [Decision Letter · Decision Letter 1]

16 Apr 2021

Time distribution for pharmacists conducting a randomized controlled trial– an observational time and motion study

PONE-D-21-02659R1

Dear Dr. Havnes,

We’re pleased to inform you that your manuscript has been judged scientifically suitable for publication and will be formally accepted for publication once it meets all outstanding technical requirements.

Kind regards,

Binaya Sapkota, PharmD

Academic Editor

PLOS ONE

Additional Editor Comments (optional):

Reviewers' comments:

Reviewer's Responses to Questions

**Comments to the Author**

1. If the authors have adequately addressed your comments raised in a previous round of review and you feel that this manuscript is now acceptable for publication, you may indicate that here to bypass the “Comments to the Author” section, enter your conflict of interest statement in the “Confidential to Editor” section, and submit your "Accept" recommendation.

Reviewer #1: (No Response)

Reviewer #2: All comments have been addressed

2. Is the manuscript technically sound, and do the data support the conclusions?

Reviewer #1: Yes

Reviewer #2: Yes

3. Has the statistical analysis been performed appropriately and rigorously? 

Reviewer #1: Yes

Reviewer #2: Yes

4. Have the authors made all data underlying the findings in their manuscript fully available?

Reviewer #1: Yes

Reviewer #2: Yes

5. Is the manuscript presented in an intelligible fashion and written in standard English?

Reviewer #1: Yes

Reviewer #2: Yes

6. Review Comments to the Author

Reviewer #1: Thank you for addressing my comments. The referencing has not been corrected to correspond with Vancouver referencing style but I have left this with the Editor.

Reviewer #2: The authors have made the revision according to the comments provided earlier. I am satisfied with the quality of the revised ms.

7. PLOS authors have the option to publish the peer review history of their article (what does this mean?). If published, this will include your full peer review and any attached files.

Reviewer #1: **Yes: **Judith Singleton

Reviewer #2: **Yes: **Mohamed Izham Mohamed Ibrahim

---

## [Editor Report · Acceptance letter]

23 Apr 2021

PONE-D-21-02659R1 

Time distribution for pharmacists conducting a randomized controlled trial– an observational time and motion study 

Dear Dr. Havnes:

I'm pleased to inform you that your manuscript has been deemed suitable for publication in PLOS ONE. Congratulations! Your manuscript is now with our production department. 

Kind regards, 

on behalf of

Dr. Binaya Sapkota 

Academic Editor

PLOS ONE